# Involvement of Urm1, a Ubiquitin-Like Protein, in the Regulation of Oxidative Stress Response of *Toxoplasma gondii*

Qianqian Tan,[a] Jinwen Wang,[a] Junpeng Chen,[a] Xiaomei Liu,[a] Xiao Chen,[a] Qianqian Xiao,[a] Jinxuan Li,[a] Hongmei Li,[a,b,c]
Xiaomin Zhao,[a,b,c] Xiao Zhang[a,b,c]

[a]Department of Preventive Veterinary Medicine, College of Veterinary Medicine, Shandong Agricultural University, Tai'an, China
[b]Shandong Provincial Key Laboratory of Animal Biotechnology and Disease Control and Prevention, Shandong Agricultural University, Tai'an, China
[c]Shandong Provincial Engineering Technology Research Center of Animal Disease Control and Prevention, Shandong Agricultural University, Tai'an, China

**ABSTRACT** Ubiquitin-related modifier 1 (Urm1) is a ubiquitin-like molecule (UBL) with the ability to act as a posttranslational protein modifier. Here, we characterized the *Toxoplasma gondii* homolog of URM1 located in the tachyzoite cytoplasm. The total loss of the TgURM1 resulted in a significant reduction in parasite invasion, replication, and virulence in mice, revealing that TgURM1 plays a pivotal role in *T. gondii* survival. For TgURM1, urmylation was significantly induced by oxidative stress, and mutations of the C-terminal glycine-glycine motif of TgURM1 blocked the urmylation process. Furthermore, the TgURM1 knockout strain was intolerant to oxidative stress, suggesting that TgURM1 is involved in the oxidative stress process. TgAHP1, an alkyl hydroperoxide reductase, was screened via proximity-based protein labeling techniques and proteomics and was shown to interact with TgURM1 under oxidative stress conditions. In conclusion, TgURM1 is a UBL protein involved in the response of *Toxoplasma* to oxidative stress.

**IMPORTANCE** *T. gondii* has an intricate life cycle which involves multiple morphologically and physiologically distinct stages, and posttranslational modifications (PTMs) may be key regulators of protein expression at relevant life cycle stages. In recent years, ubiquitin-like proteins with modification functions have been discovered and studied, including Sumo, Rub1, ATG8, and ATG12. Ubiquitin-related modifier 1 (Urm1) is a ubiquitin-like molecule (UBL), which is considered to be the oldest ubiquitin-like system. In this study, we identified the Urm1 gene in *Toxoplasma* and explored that the urmylation of Urm1 was significantly induced by oxidative stress. Fewer studies have been conducted on ubiquitin-like proteins of parasites, and our results provide theoretical support for the research of metabolic regulation and antioxidative stress processes in *T. gondii*.

**KEYWORDS** *Toxoplasma gondii*, urmylation, Urm1, oxidative stress

*T*oxoplasma gondii is a protozoan parasite that infects the nucleated cells of warm-blooded vertebrates, and an estimated one-third of the world's population is infected with *T. gondii* (1). As an opportunistic pathogen (2), although most infections are asymptomatic, it can trigger several complications in immunocompromised individuals (3). In addition, infection during pregnancy can cause serious damage, such as miscarriage and stillbirth (3, 4). Severe symptoms are observed in newborns, including blindness, mental retardation, and hydrocephalus when the primoinfection occurs in the pregnant woman (5).

*T. gondii* is a protozoan parasite with a complex life cycle and undergoes multiple morphologies (6), including tachyzoites, bradyzoites, schizonts, etc. The entire cycle requires complex regulation of gene expression, which depends on transcriptional control mechanisms, translation, posttranslational modifications (PTMs), and protein degradation (7). Various PTMs have been found in protozoans, including phosphorylation, acetylation, palmitoylation, methylation, glycosylation, and ubiquitination (8–10). In recent years, ubiquitin-like proteins with similar modification functions have been successively discovered and

Address correspondence to Xiaomin Zhao, xmzhao66@163.com, or Xiao Zhang, tzhangxiao@126.com.

The authors declare no conflict of interest.

studied, including Sumo, Rub1, ATG8, and ATG12 (11). However, fewer studies have been conducted on ubiquitin-like proteins of parasites.

Ubiquitin-related modifier 1 (Urm1) is a ubiquitin-like molecule (UBL), which is considered to be the oldest ubiquitin-like system, mainly due to its dual function. URM1 was first identified as a UBL protein in *Saccharomyces cerevisiae*. Under the action of Uba4 (similar to the function of E1 activase), URM1 is covalently coupled to a variety of target proteins in yeast and performs different functions (12). Further studies revealed that URM1 is associated with prokaryotes and has similar sequences to the biopterin carrier protein MoaD (a small subunit of methotrexate [MPT] synthase) and ThiS, which are involved in the biosynthesis of methotrexate and thiamine, respectively. Subsequently, Urm1 was shown to have sulfur transfer activity similar to that of its prokaryotic ancestor (13, 14).

In our study, the TgURM1 knockout strain was intolerant to oxidative stress, suggesting that TgURM1 may be involved in the oxidative stress process. *T. gondii* is sensitive to both endogenous and exogenous oxidative stress, and minor changes in redox balance can lead to the disruption of *T. gondii* oxidative homeostasis and, ultimately, its death (15). To protect *T. gondii* against dangerous reactive oxygen species (ROS), multiple response systems have evolved, including superoxide dismutases (SODs), catalase (CAT), and peroxiredoxins (15–17). SOD mainly catalyzes $O_2\cdot^-$ into $H_2O_2$ and oxygen (18). CAT acts downstream of SOD, which catalyzes the conversion of $H_2O_2$ into molecules of water and oxygen (19) Peroxiredoxin (Prx) is an antioxidant protein family, which consists of cysteine-based peroxidases that play a central role in keeping the $H_2O_2$ at physiological levels that widely exist in organisms (20), and classical antioxidant systems such as thioredoxins (Trx) and GSH have also been found in *Toxoplasma* (21). The oxidative stress response systems involved in TgURM1 regulation require further confirmation.

In this study, we identified the Urm1 gene in *Toxoplasma* and explored the role of TgURM1 in oxidative stress. Our results suggested that urmylation of TgURM1 was significantly induced by oxidative stress, and TgAHP1 is an interacting protein of TgUrm1 involved in the antioxidant stress process.

## RESULTS

**Characterization of URM1 in *T. gondii*.** The *T. gondii* genome contains a unique Urm1 homolog, encoded by the unannotated gene TGGT1_233200 (ToxoDB database). Sequence alignment analysis of Urm1 homologs in different species showed that TGGT1_233200 displayed all the features of a typical UBL, including a predicted $\beta$-grasp structural fold and a conserved C-terminal diglycine (GG) motif (Fig. 1A), and we here refer to TGGT1_233200 as TgUrm1. Phylogenetic analysis places TgUrm1 together with other Urm1 homologous genes, and a high homology was observed in *Caenorhabditis elegans*. In addition, TgUrm1 was also homologous to MoaD and ThiS in *Escherichia coli*, which indicated an evolutionary ancestor being shared with the sulfur carriers MoaD and ThiS (Fig. 1B). The hemagglutinin (HA)-TgURM1 strain in which the 3xHA was tagged at TgUrm1's N terminus was used to observe the expression of Urm1 in *T. gondii*, and Western blot analysis showed that the TgUrm1 was expressed successfully (Fig. 1C). The localization of TgURM1 in *T. gondii* was determined using an immunofluorescence assay, with the results indicating that TgURM1 is a protein that primarily resides in the cytoplasm of the parasite (Fig. 1D).

**Parasites lacking URM1 with significantly reduced invasion and proliferation.** To further investigate the function of URM1, we generated a monoclonal mutant of the entire URM1 locus that was deleted by CRISPR/Cas9-mediated homology-directed repair (HDR). The knockout strategy involved gene insertion of a modified allele of the gene encoding dihydrofolate reductase (DHFR), which confers resistance to pyrimethamine, into the entire TgUrm1 locus (Fig. 2A), with clones being generated and identified by PCR (Fig. 2B). To test the viability of the ΔUrm1 strain, a plaque assay was performed on the DF-1 monolayer, and the ΔUrm1 strain showed a significant reduction in plaque quantity and area compared to the wild-type RHΔKu80 strain and ComΔUrm1, indicating a growth defect of the ΔUrm1 strain (Fig. 2C).

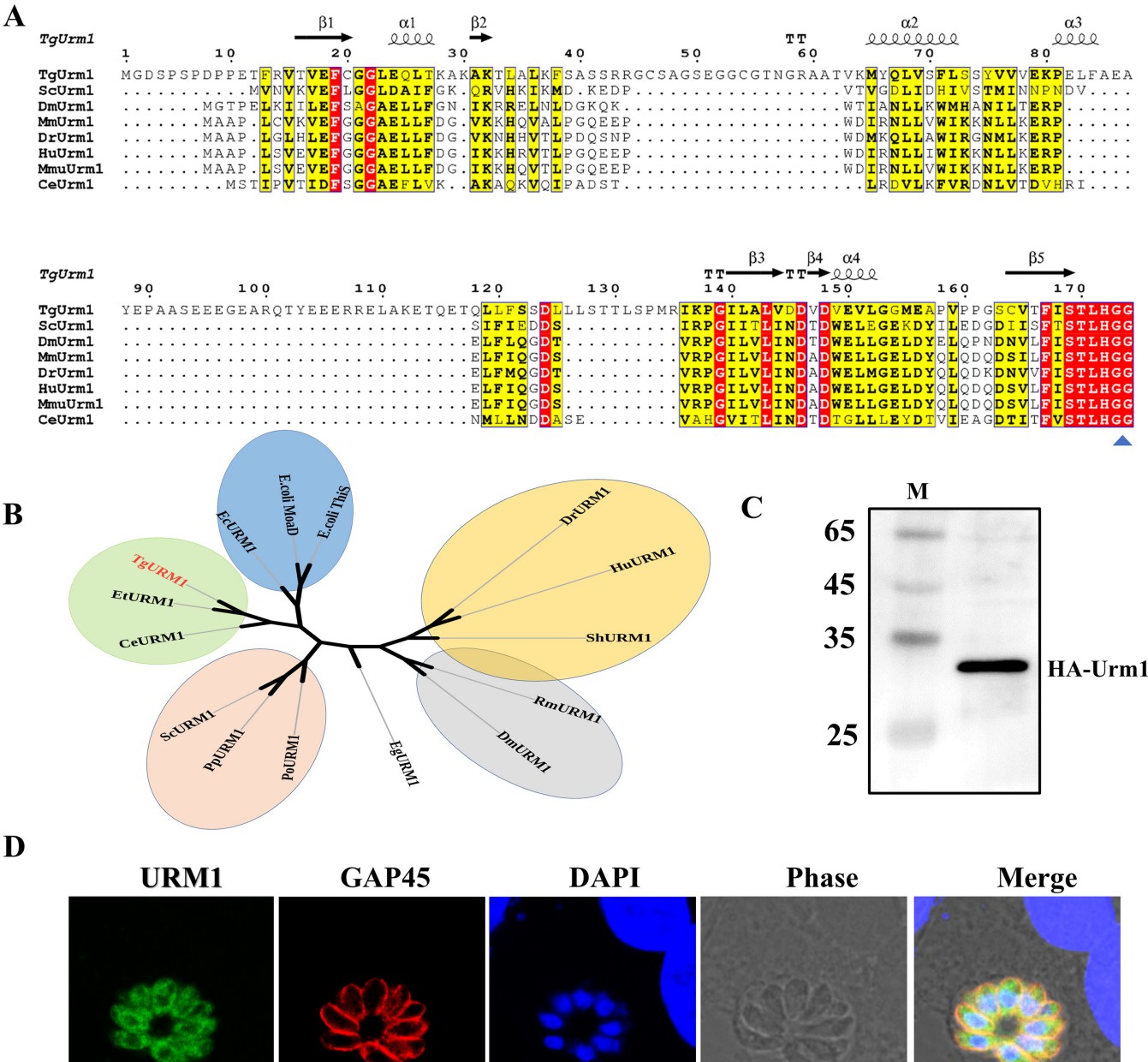

**FIG 1** The *Toxoplasma* Urm1 homolog, TGGT1_233200, is an evolutionary conserved UBL localized in the cytoplasm of *T. gondii*. (A) Alignment of amino acid sequences of URM1 from *T. gondii* and other model organisms. Sequence alignment of TgUrm1 (TGGT1_233200) with homologous counterparts in *Saccharomyces cerevisiae* S288C, *Drosophila melanogaster*, *Mus musculus*, *Danio rerio*, *Homo sapiens*, *Macaca mulatta*, and *Caenorhabditis elegans*. Amino acid sequences are aligned with ClustalW. Yellow- and red-filled rectangular frames show conserved amino acids; dots indicate gaps or missing residues. (B) Phylogenetic tree of URM1 proteins in several species. These species include *Toxoplasma*, *Encephalitozoon cuniculi*, *Drosophila melanogaster*, *Homo sapiens*, *Saccharomyces cerevisiae*, *Plasmodium ovale*, *Parasitella parasitica*, *Eimeria tenella*, *Danio rerio*, *Caenorhabditis elegans*, *E. coli ThiS*, *E. coli MoaD*, *Schistosoma haematobium*, *Echinococcus granulosus*, and *Rhipicephalus microplus*. The phylogenetic tree was constructed by neighbor-joining algorithm and displayed via DNAMAN. (C) The expression of HA-Urm1 fusions was checked by Western blotting in transgenic parasites, and 28 kDa molecular weight of HA-TgURM1 was observed. (D) IFA used for TgURM1 localization observation. TgURM1 (green) and TgGAP45 (red) in RHΔKu80 determined the localization. DAPI, 4′,6-diamidino-2-phenylindole, nuclear dye; TgGAP45, gliding-associated protein 45; BF, brightfield. Scale, 2.5 μm.

The growth defect was further confirmed, and the attenuated invasion and proliferation trends were observed in the ΔUrm1 strain compared to wild-type RHΔKu80 strain and ComΔUrm1 strain (Fig. 2D and E). Gliding and egress of parasites were evaluated, and the results showed that the parasites were defective in gliding, with less obvious circular gliding, upright twirling, and helical rotation path (Fig. 2F and G), but without significant differences in the egress process (Fig. 2H). In conclusion, loss of URM1 in *T. gondii* leads to the reduced gliding, invasion, and intracellular proliferation of parasites.

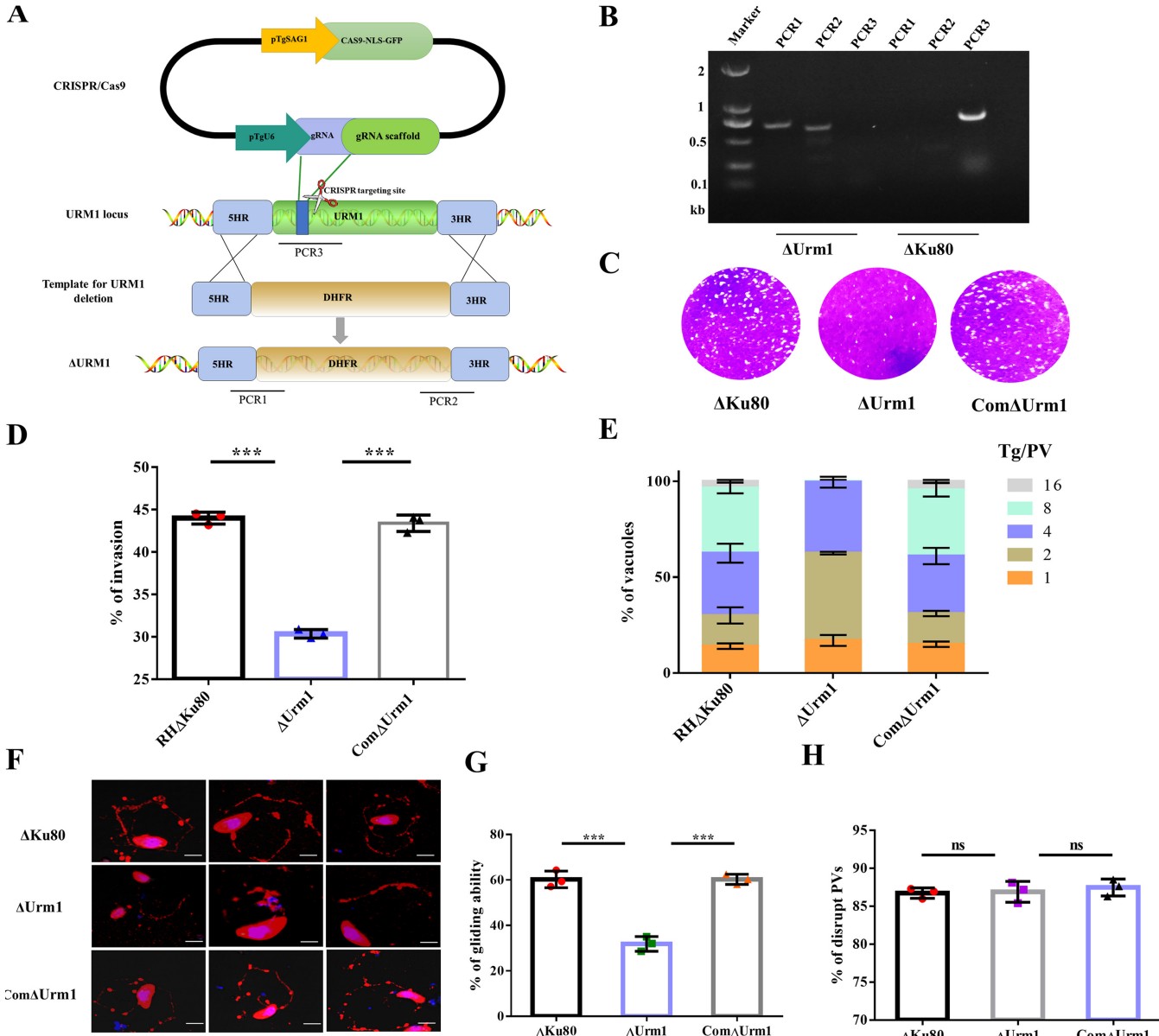

**FIG 2** Characterization of the Urm1 deletion strain. (A) Schematic diagram of Urm1 knockout in RHΔKu80 by CRISPR/Cas9-mediated homologous gene replacement. PCR1 and PCR2 check the 5′ and 3′ integration of the selection marker, whereas PCR3 examines the successful deletion of Urm1 genes. (B) Diagnostic PCRs for ΔUrm1 mutant. RHΔKu80 strain was included as control. (C) Seven-day plaque assays were performed on ΔUrm1, ComΔUrm1, and RHΔKu80. (D) Differential invasion efficiency of RHΔKu80, ComΔUrm1, and ΔUrm1 strains. Each with three replicates. ***, $P < 0.001$, Student's *t* test. (E) Intracellular replication assay for evaluating the proliferation of RHΔKu80, ComΔUrm1, and ΔUrm1 strains *in vitro*. (F) Gliding assay of ΔUrm1, ComΔUrm1, and RHΔKu80. Gliding was observed using IFA as the surface protein SAG1, which was left behind with the gliding of the parasites. Scale, 2.5 $\mu$m. (G) Statistics of the number of tachyzoites having the ability of gliding in RHΔKu80 strain, ComΔUrm1 strain, and ΔUrm1 strain. Parasites that could not complete one of the three gliding modes were defined as gliding defect. ***, $P < 0.001$, Student's *t* test. (H) Egress capacity of the ΔUrm1 strain. Compared to ΔKu80 and ComΔUrm1, ΔUrm1 showed a near-egress rate. ns, $P > 0.05$, indicates not significant, Student's *t* test.

**Loss of URM1 resulting in reduced virulence in mice.** To investigate the virulence of the ΔUrm1 strain in mice, mice were injected intraperitoneally with 500 tachyzoites of ΔUrm1, ComΔUrm1, Urm1-OE (TgURM1 overexpressed in the knockout strain), and ΔKu80, respectively. Compared with the other three strains, the virulence of the ΔUrm1 strain was significantly attenuated (Fig. 3A). At 30 days postinfection, all surviving mice were sacrificed, and sera were tested. All samples were seropositive, indicating successful parasite infection. To further evaluate the virulence of the ΔUrm1 strain in mice, BALB/c mice were infected with increasing and decreasing numbers of tachyzoites, and all mice died with high numbers of ΔUrm1 strain infections (Fig. 3B). Variance analysis showed that knockout of Urm1

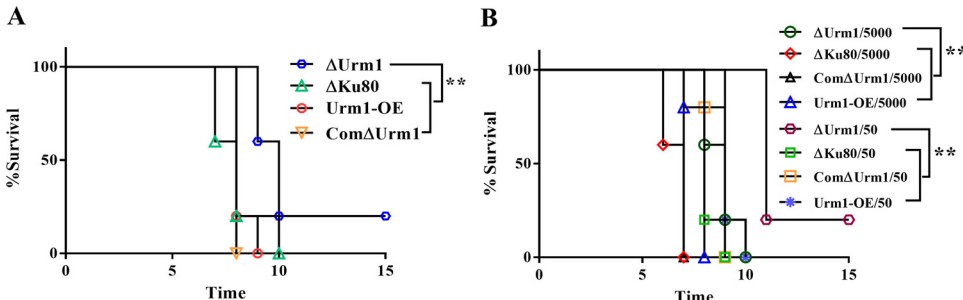

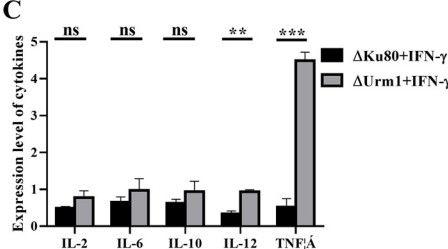

**FIG 3** Loss of Urm1 resulting in reduced virulence in mice. (A) Female BALB/c mice (5 per group) were infected with 500 of RHΔKu80, Urm1-OE, ComΔUrm1, and ΔUrm1 tachyzoites by intraperitoneal injection as indicated. Infected mice were monitored at least 3 times daily. **, $P < 0.01$, Gehan-Breslow-Wilcoxon tests. (B) Survival curve of mice infected with the designated strains. RHΔKu80, Urm1-OE, ComΔUrm1, and ΔUrm1 strains were used to infect BALB/c mice (5,000/50 tachyzoites per mouse, $n = 5$ mice per strain), and the survival conditions of mice were recorded daily. **, $P < 0.01$, Gehan-Breslow-Wilcoxon tests. (C) Analysis of cytokines in parasite-infected macrophages of mice. Values shown are means ± SEM from three independent experiments ($n = 3$), each with three replicates. ns, $P > 0.05$; **, $P < 0.01$; ***, $P < 0.001$, Student's $t$ test.

significantly reduced the pathogenicity of *T. gondii* in the two infection dose groups. Inflammatory cytokines were detected after *T. gondii* infected macrophages of mice for 24 h. The levels of interleukin 12 (IL-12) and tumor necrosis factor alpha (TNF-$\alpha$), key cytokines that inhibit *Toxoplasma* infection, significantly increased in ΔUrm1 strain-infecting cells (Fig. 3C). These indicated that the knockout strain defected in immune evasion and was easily eliminated by the mouse immune system.

**Oxidative stress induces the conjugation of URM1 *in vivo*.** Ubiquitin-like proteins (UBLs) play important roles in cell biology by covalently conjugating to target proteins. To investigate the conjugated proteins of TgURM1, HEK293T cells were stably transduced with FLAG epitope-tagged TgURM1 and, in addition, indicated Urm1 as a protein primarily residing in the cytoplasm (Fig. 4A). Cytoplasmic separation was made in URM1-transfected HEK293T cells, and Western blotting was performed to confirm the cytoplasmic localization of URM1 (Fig. 4B). The transduced cells were treated with a variety of stressors, and a distinct pattern of higher-molecular-weight polypeptides was observed in the group treated with the oxidative stressor diamide (Dia) (Fig. 4C). The Urm1 endogenous marker strain expressing HA tags fused at the N terminus was treated with different oxidants, and the similar ubiquitination of URM1 in *T. gondii* was also observed, indicating that TgURM1 may be involved in the antioxidative stress response of *T. gondii* (Fig. 4D). Some studies have been reported the reaction that oxidative stress induces conjugation of Urm1 to target proteins requires the C terminus of Urm1 in both *S. cerevisiae* and mammalian cells (22). Truncated TgUrm1 (ΔGG) with a FLAG epitope tag was transduced into HEK293T cells to test whether conjugate formation requires the C-terminal diglycine motif, and fewer URM1 conjugations were formed under oxidative stress diamide treatment conditions (Fig. 4E), indicating that the C terminus is crucial for conjugation of Urm1 to target proteins. Thus, oxidative stress enhances the conjugation of URM1 to target proteins in *T. gondii*, and oxidant-induced urmylation is dependent on the C-terminal glycine of Urm1 in *T. gondii*.

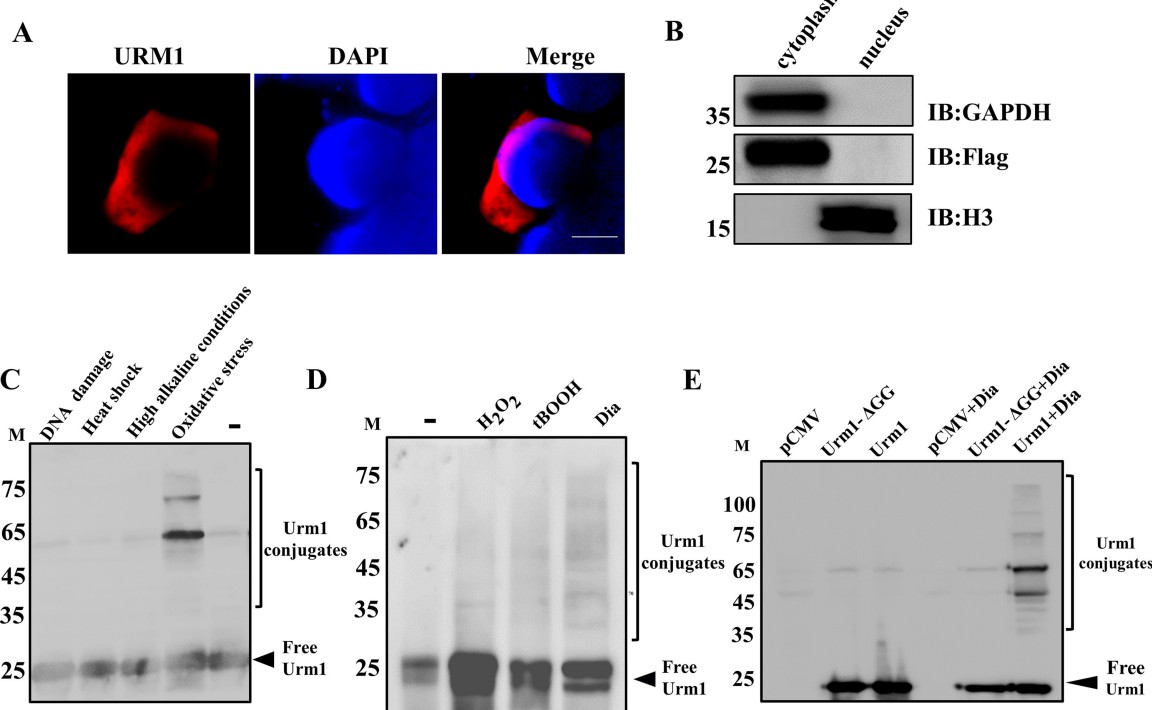

**FIG 4** Oxidative stress induces the conjugation of URM1 *in vivo*. (A) TgURM1 expressed in cytoplasm of transfected HEK293T cells. Scale, 5 $\mu$m. (B) Western blotting was performed to confirm the cytoplasmic localization of URM1. GAPDH (glyceraldehyde-3-phosphate dehydrogenase), cytoplasmic reference; H3, nuclear reference. (C) FLAG-Urm1-transduced HEK293T cells were treated with different stressors and lysed in 1% SDS. Total cell lysates were resolved by SDS/PAGE and subjected to anti-FLAG immunoblotting. The treatment is as follows: 400 $\mu$M diamide for 10 min, DNA damage caused by 24.5 Gy of gamma irradiation followed by a 4-h recovery period, heat shock by incubation at 40°C for 6 h, and high alkaline environment by incubation at pH 8.2 medium for 6 h. (D) The HA-Urm1 strain was treated with different oxidants. The treatment is as follows: 400 $\mu$M diamide, $H_2O_2$ or nitrite for 10 min, $10^{-4}$ M tBOOH for 2 h. (E) FLAG-Urm1 and FLAG-Urm1 $\Delta$GG-transduced HEK293T cells were treated with 400 $\mu$M diamide (Dia) for 10 min and lysed in 1% SDS. Total cell lysates were resolved by SDS/PAGE and subjected to anti-FLAG immunoblotting.

**Screening for URM1-interacting proteins.** Oxidative stress-induced conjugated proteins of TgURM1 were screened using biotin-adjacent labeling and proteomics. The TurboID-HA-Urm1 construct was cotransfected with the uracil phosphoribosyltransferase (UPRT) targeting CRISPR plasmid into $\Delta$Urm1 to generate transgenic strains expressing TurboID-HA-Urm1 fusions from the UPRT locus (Fig. 5A), with clones being generated and identified by PCR (Fig. 5B). By probing with anti-HA, TurboID-HA-Urm1 fusion was efficiently expressed (Fig. 5C), and indirect immunofluorescence showed that a significant number of interacting proteins were labeled with biotin (Fig. 5D). In the presence or absence of Dia, biotinylated proteins were enriched from parasite lysates by streptavidin-conjugated beads, which were confirmed by Western blotting (Fig. 5E) and analyzed by mass spectrometry for identification. A total of 51 proteins were identified using biotin-adjacent labeling method in the presence of Dia, while 107 proteins were identified by proteomics. After comprehensive analysis of all identification, 22 promising proteins were selected (Fig. 5F), which involved regulating oxidative stress response, immune evasion, and metabolism.

**AHP1 is targeted by URM1 urmylation in response to oxidative stress.** TgAHP1 (TGGT1_286630) was first reported in our previous study (23), which was identified as a peroxisomal protein with cysteine-dependent peroxidase activity in *T. gondii*. Interestingly, TgAHP1 was found in the mass spectrometry data (Fig. 5E), indicating that AHP1-URM1 complexes may be detected in the presence of oxidant. Initially, we validated the binding of two proteins in the yeast two-hybrid system, and no yeast survived in SD-4 culture medium without oxidant (Fig. 6A). Thus, we conducted coimmunoprecipitation to explore whether the oxidant could affect the interaction between two proteins. In line with our assumption, the oxidant remarkably contributed to the

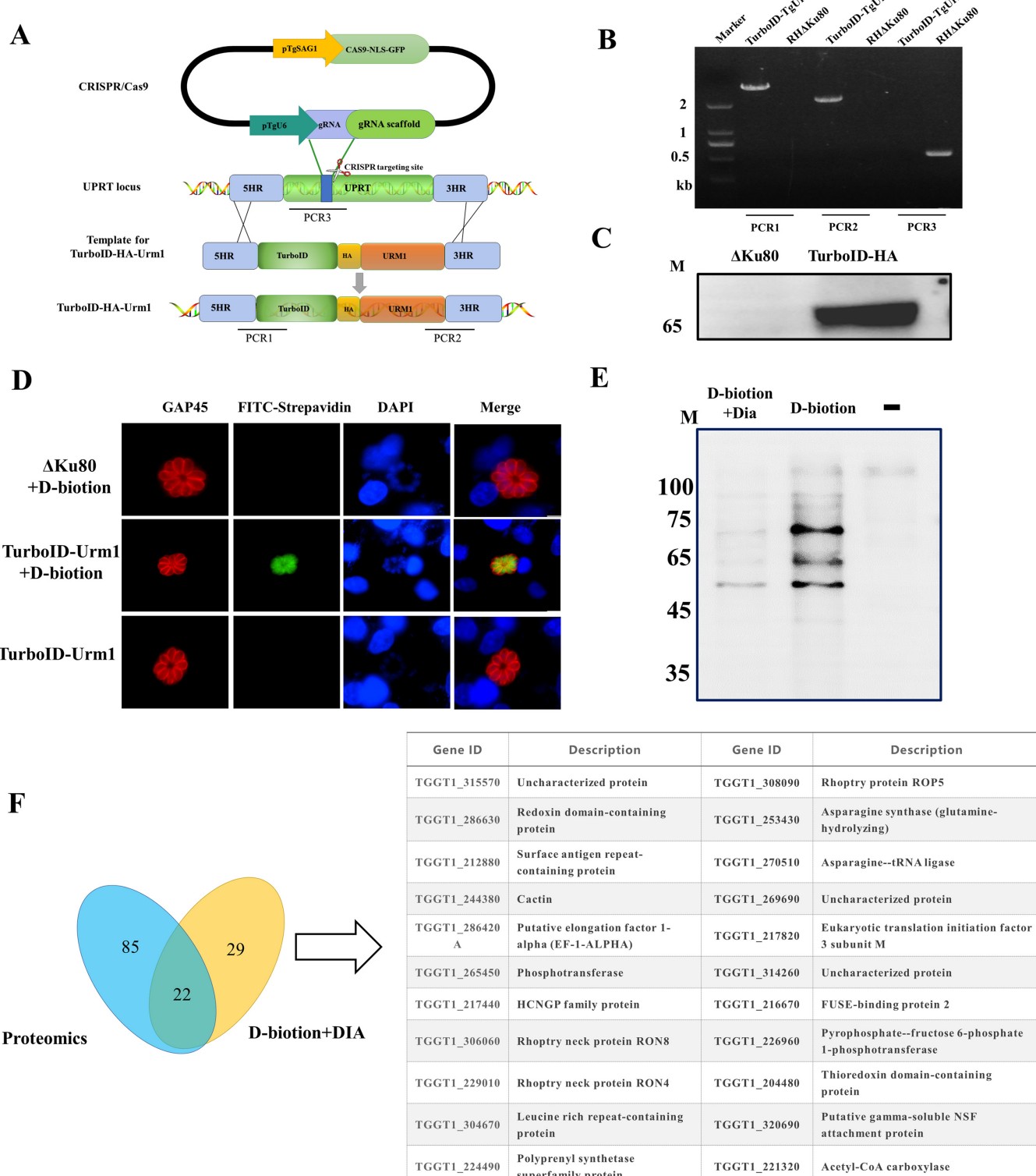

**FIG 5** TurboID-HA-Urm1-directed protein biotinylation. (A) Schematic diagram of the CRISPR/CAS9-mediated TurboID-HA-Urm1 insertion at the UPRT locus. PCR1 and PCR2 detected the correct insertion sites, and PCR3 detected the UPRT endogenous locus. (B) Diagnostic PCR demonstrating homologous integration and gene disruption in TurboID-HA-Urm1 strain compared with the parental line RHΔKu80. (C) Immunoblot assays checking the expression of TurboID-HA-Urm1 fusions in transgenic parasites. (D) Visualization of biotinylated proteins in parasites. TurboID-HA-Urm1 tachyzoites were grown on DF-1 monolayers in the presence (+) or absence (−) of biotin treatments and analyzed by indirect immunofluorescence staining. Biotinylated proteins were detected by FITC-conjugated streptavidin. Scale, 2.5 μm. (E) Enriched and purified biotin proteins were detected by immunoblot. TurboID-HA-Urm1 tachyzoites were grown on Vero in the presence (+) or absence (−) of biotin treatments. Biotinylated proteins were detected by horseradish peroxidase (HRP)-conjugated streptavidin. (F) Strategies for mass spectrometry data analysis.

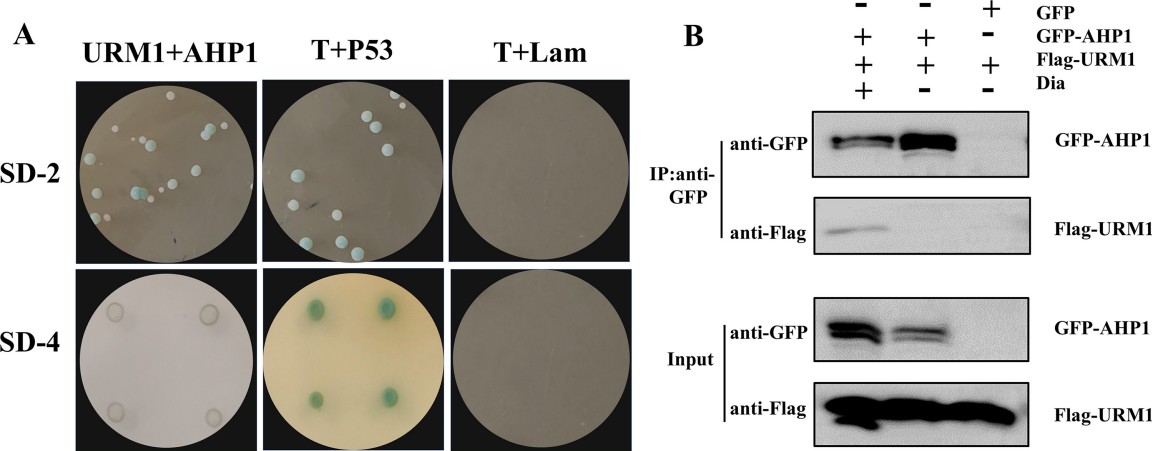

**FIG 6** AHP1 is targeted by urmylation in response to oxidative stress. (A) Analysis of the interaction between URM1 and AHP1 by yeast two-hybrid system. Photographs were taken after 5 days of yeast cell growth on −Leu/−Trp (SD-2) medium or −Leu/−Trp/−His/−adenine (SD-4) medium. T+Lam and T+p53 were used as negative and positive controls for the yeast two-hybrid system, respectively. (B) Coimmunoprecipitation of TgAHP1 with TgURM1. The interaction of TgURM1 and TgAHP1 was validated by coimmunoprecipitation in HEK293. GFP-tagged TgAHP1 and FLAG-tagged TgURM1 were coexpressed in HEK293T. Protein extracts were incubated with (+) or without (−) 400 $\mu$M Dia for 10 min prior to immunoprecipitation using GFP magnetic beads.

AHP1-URM1 interaction in cotransfection HEK293 cells. Lacking oxidant treatment, the interaction between AHP1-URM1 was invisible, indicating which oxidant was an inducer of AHP1-URM1 complexes.

**Loss of URM1 causes intolerance of *Toxoplasma* to oxidative stress.** In our previous study, we have shown that the ΔAhp1 strain was sensitive to the oxidizing agent *tert*-butyl hydroperoxide (tBOOH) (23), and the oxidant was an inducer of AHP1-URM1 complexes (Fig. 6B). Thus, TgURM1 transduced cells treated with tBOOH to confirm oxidative stress-induced urmylation, and results showed a significant ladder of conjugated proteins (Fig. 7A). Invasion and virulence assays were performed to explore whether Dia and tBOOH could affect the survival of the ΔUrm1 strain. In line with our assumption, the statistical results showed that the ΔUrm1 strain was intolerant to oxidants than the wild-type (WT) strain and ComΔUrm1 strain (Fig. 7B to E). Likewise, the knockout strain also had an attenuated virulence in mice in the presence of oxidants (Fig. 7F).

To investigate ROS level in ΔUrm1 strain treated with tBOOH or Dia, tachyzoites of the ΔUrm1, WT, and ComΔUrm1 strains were labeled with 2,7-dichlorodihydrofluorescein diacetate (DCFH-DA) and measured with fluorescence intensity of 4',6-dicyanoflavan (DCF). The results showed that the ROS significantly accumulated more in the ΔUrm1 strain than WT and ComΔUrm1 strains (Fig. 7G), which may be the main cause of parasites' decreased pathogenicity and viability.

## DISCUSSION

UBL Urm1 was first identified in *T. gondii*, and oxidative stress was found to be a strong inducer of TgURM1 conjugation. Under oxidative stress, total loss of TgUrm1 results in parasites' significantly reduced invasion, intracellular proliferation, and pathogenicity in mice, which indicated that URM1 is involved in the oxidative stress response of *T. gondii*.

Conjugation of Urm1 to target proteins, termed urmylation, is highly conserved during evolution. Ubiquitination has been shown to rely on the Urm1-specific E1 enzyme ubiquitin-like modifier activator 4 (Uba4) in yeast (12), known in mammals as molybdenum cofactor synthesis 3 (MOCS3) (24). Uba4 in yeast has been identified as the designated E1 enzyme of Urm1, which differs from all other UBL E1 enzymes because it contains a C-terminal RHD in addition to its N-terminal E1-like domain (25). Studies have found that the carboxyl group of the C-terminal glycine of ScUrm1 forms a thioester

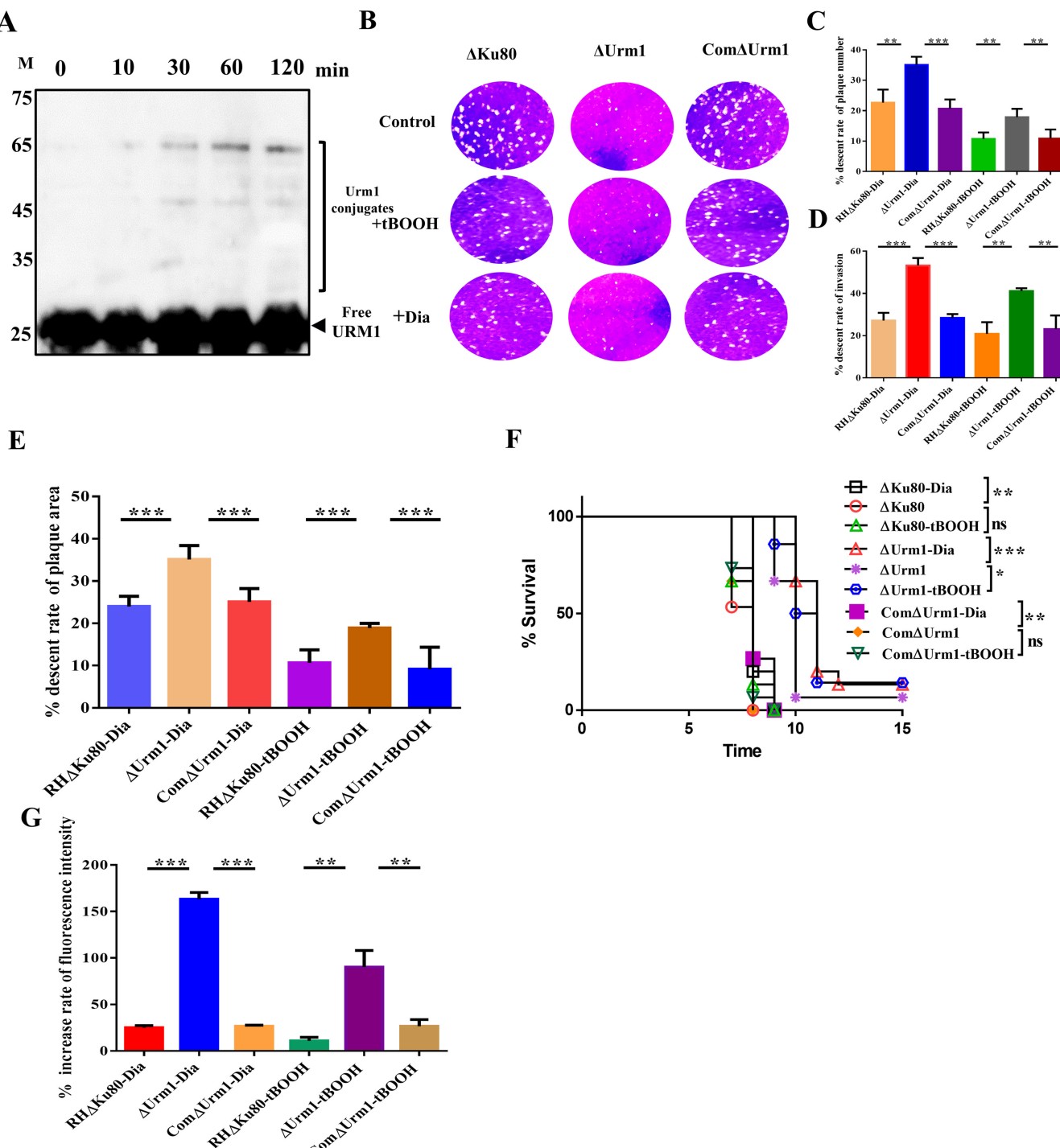

**FIG 7** Loss of URM1 causes intolerance of *Toxoplasma* to oxidative stress. (A) The urmylation induced by $10^{-4}$ M tBOOH with different stimulation times. (B) Seven-day plaque assay conducted on ΔUrm1, ComΔUrm1, and RHΔKu80 strains after stimulation with $10^{-4}$ M tBOOH or 400 $\mu$M Dia. (C and D) Descent rate of the number and area of the plaque assay. **, $P < 0.01$; ***, $P < 0.001$, Student's $t$ test. (E) Descent invasion rate of RHΔKu80, ComΔUrm1, and ΔUrm1 strains with oxidant stimulation. **, $P < 0.01$; ***, $P < 0.001$, Student's $t$ test. (F) Survival curve of mice infected with the designated strains. RHΔKu80, ComΔUrm1, and ΔUrm1 strains stimulated with $10^{-4}$ M tBOOH were infected with BALB/c mice by intraperitoneal injection (500 tachyzoites per mouse, $n = 15$ mice per strain), and the survival conditions of mice were recorded daily. *, $P < 0.05$; **, $P < 0.01$; ***, $P < 0.001$, Gehan-Breslow-Wilcoxon tests. (G) Increased rate of ROS in ΔUrm1, ComΔUrm1, and WT strain post-oxidant stimulation. The formula for the ROS increase rate is simplified to the following: increase rate = [OX(+) − OX(−)]/OX(−) × 100%, where OX(+) is the ROS levels of oxidant treatment group, and OX(−) is the ROS levels of oxidant nontreatment group. **, $P < 0.01$; ***, $P < 0.001$, Student's $t$ test.

with the typical cysteine, which is located in the cross-loop of the Uba4 E1 domain (22). ScUrm1 then formed an acyl-persulfide bond with cysteines in the RHD. *In vivo* regeneration of ScUba4 is accompanied by the release of thiocarboxylated ScUrm1 (25). Strikingly, no Urm1-specific E2 and E3 enzymes have been identified to date. Our laboratory has also identified a UBA4 homolog in *T. gondii* and demonstrated that TgURM1 interacts with TgUBA4 in a specific domain. However, urmylation of TgURM1 was not completely dependent on the sulfide modification of TgUBA4 (X. Zhang, unpublished data). Thus, the crucial role of TgUBA4 in TgURM1 urmylation is still unknown.

Urm1 shares certain notable features with ubiquitin, namely, the characteristic five-stranded $\beta$-grasp fold and contains a C-terminal glycine (Gly) or diglycine (Gly-Gly) motif that is required for activation and conjugation to substrates (26). Some studies have shown that ScUrm1 is covalently conjugated to lysine residues of target proteins in a manner similar to ubiquitylation (22). The distinct pattern of higher-molecular-weight polypeptides observed with Dia treatment shows that TgURM1 covalently conjugates to target proteins. However, immunoprecipitation revealed a solitary molecule of TgAHP1 rather than TgURM1-AHP1 conjugation, indicating the formation of noncovalent TgURM1-AHP1 conjugation. Some reports have suggested that besides covalent linkages, UBLs interact noncovalently with proteins that depend on ubiquitin-binding domains other than conjugation sites (typically conjugated ubiquitin) in *T. gondii* (27). The integration mode of TgURM1 with target proteins follows the traditional lysine-directed covalent binding or noncovalent combination, as described, which is still unclear in *Toxoplasma*.

The increased levels of urmylation and altered sensitivity were observed during oxidative stress in a variety of model systems, including yeast, mammalian cells, and *Drosophila melanogaster* (22, 28, 29). The appearance of multiple high-molecular-weight bands, recognized in lysates treated with Dia, has been able to demonstrate the participation of TgURM1 in the resistance to oxidative stress in *T. gondii*. This is emphasized by the results of TgURM1 conjugation to the target protein TgAHP1, which is a peroxisomal protein with cysteine-dependent peroxidase activity and sensitivity to tBOOH. TgAHP1 is a member of the Prxs family, which follows a cysteine-dependent redox process, and Cys[166] was certified to be a key enzymatic active site, as shown in our previous study (23). ScURM1 was reported to form a linkage based on the lysine sites of target proteins, and the lysine site of TgAHP1 was also found to be in close proximity to the enzymatic active site (30). Whether the TgURM1-AHP1-binding sites are associated with the adjacent Cys[166] enzyme site of TgAHP1 to regulate the oxidative stress response deserves further study.

Subtly, the adjacent biotin-labeling technique was used to screen for proteins that interact with TgURM1, and a series of proteins were identified, including RON4 and ROP5. RON4, a protein recently described to be localized to the duct-like rhoptry necks of *T. gondii* tachyzoites, is closely related to the formation of the motile host cell complex (MJ) during the parasite invasion process (31). ROP5 controls pathogenicity of parasites by blocking interferon gamma (IFN-$\gamma$)-mediated clearance in activated macrophages (32). In low-dose infection, surviving mice were observed, and the decrease in the expression of immune evasion factors may be an important reason for the weakened pathogenicity of the ΔUrm1 strain of mice. Several studies have shown that Urm1 plays an important role in regulating the expression of genes involved in metabolism. Lack of URM1 causes derepression of the GAP1 gene and simultaneously inhibits the expression of CIT2, which is a tricarboxylic acid (TCA) cycle gene involved in the production of glutamate and glutamine (33). Our study revealed a low proliferation rate of the ΔUrm1 strain in cells, speculating that TgURM1 may participate in certain metabolic pathways to determine the growth of *Toxoplasma*.

## MATERIALS AND METHODS

**Parasite strain, cell, and mice.** The type I strain RH ΔKu80 was used for the construction of the transgenic strain, which was maintained in DF-1 cells (a spontaneously immortalized chicken embryo fibroblast cell line) or Vero cells (African green monkey kidney cells) at 37°C with 5% $CO_2$. Six-week-old female BALB/c mice were purchased from Jinan Pengyue Experimental Animal Breeding Co., Ltd., and maintained under standard conditions in accordance with the regulations specified by the Administration of Affairs

Concerning Experimental Animals. The mice used in this study were approved by the Ethics Committee for Animal Experiments of the Laboratory Animal Center of Shandong Agricultural University, China.

**Generation of Urm1 knockout and complemented strains.** To disrupt Urm1 in RHΔKu80, the sgUrm1 CRISPR plasmid was electroporated with an amplicon containing the Urm1 homology region surrounding a pyrimidine-resistant DHFR cassette. Pyrimethamine (1 $\mu$M) was used to screen for stable resistant clones, and the correct integration of DHFR into the Urm1 locus was verified by PCR as described by Bang Shen et al. (34). To construct complementary strains, the 3×HA::RHΔKu80 Urm1 and the sgUrm1 CRISPR plasmid were coelectroporated and integrated into the Urm1 locus to replace the original TgUrm1. Monoclonal strain was screened by immunofluorescence using monoclonal antibody (MAb) HA. Positive clones were further analyzed by PCR for correct integration at the Urm1 locus and by Western blotting for Urm1 expression.

**Plaque assay.** For the plaque assay, DF-1 monolayers growing on 12-well plates were infected with tachyzoites (500/well) of different strains. After 7 days of infection, infected cell cultures were washed with phosphate-buffered saline (PBS) and fixed with 4% paraformaldehyde, followed by staining with 2% crystal violet for 30 min to visualize plaques formed by proliferating tachyzoites (35).

**Parasite invasion assay.** A total of $1 \times 10^5$ freshly released tachyzoites of different strains were allowed to invade the DF-1 monolayer for 30 min, and then the cells were washed with PBS three times. Invaded (extracellular) versus noninvaded (intracellular) parasites were distinguished by a two-color staining protocol. Extracellular parasites were stained with mouse anti-TgSAG1, and total parasites were stained with rabbit anti-TgGAP45. Fluorescein isothiocyanate (FITC)-conjugated goat anti-mouse IgG (H+L) and Cy3-conjugated goat anti-rabbit IgG (H+L) were secondary antibodies and were incubated together. The invasion efficiency was calculated by counting the ratio of the number of infected tachyzoites/total host cells in several random fields under a fluorescence microscope (36). All strains were tested 3 times independently.

**Parasite replication assay.** Freshly isolated parasites ($1 \times 10^5$) were inoculated on DF-1 in 12-well plates. After 30 min, the extracellular parasites were removed by washing with PBS. At 24 h postinfection, indirect immunofluorescence (IFA) was performed to visualize the parasites. Following fluorescence staining, samples were examined by fluorescence microscopy to determine the number of parasites in each parasitophorous vacuole. The number of parasites in 100 randomly selected vacuoles was counted for each sample in parasite replication assay (37). All samples were anonymized prior to parasite counting. Real-time images were obtained through a microscope in three independent experiments, with two observers responsible for statistics and calculations.

**Parasite gliding assay.** Coverslips were coated by incubation in 50% fetal bovine serum diluted with PBS overnight at 4°C followed by rinsing in PBS. Freshly egressed tachyzoites were purified and resuspended in extracellular buffer (142 mM NaCl, 5 mM KCl, 1.8 mM CaCl$_2$, 1 mM MgCl$_2$, 5.6 mM glucose, and 25 mM HEPES, pH 7.4) before analysis. Parasites were added to coverslips and incubated at 37°C for 30 min and fixed with 4% formaldehyde, and IFA, using the anti-SAG1 antibody, was performed to visualize the trails. Parasites that could not complete one of the three gliding modes were defined as gliding defect (38). For all treatments, at least 50 parasite trails were measured on each experiment.

**Parasite egress assay.** Confluent DF-1 monolayers were infected with $10^5$/well purified tachyzoites and cultured at 37°C for 24 h. Intracellular parasites were then treated with 10 $\mu$M calcium ionophore A23187 (Sigma, USA) at 37°C for 5 min. Egress efficiency was determined by dividing the number of egressed vacuoles by the number of total vacuoles as previously described (39).

***Toxoplasma* infection in mice.** Six-week-old female BALB/c mice were allowed to acclimate in our facility for 1 week. The mice were randomized into several groups ($n = 5$). WT, ComΔUrm1, and ΔUrm1 tachyzoites were prepared in sterile PBS and injected intraperitoneally into the mice, which were then monitored at least 3 times a day.

**Protein biotinylation and affinity purification.** The TurboID-HA-Urm1 strain was used to infect the Vero cells for about 24 h. The culture medium was then changed to medium containing 150 $\mu$M biotin and grown at 37°C for 24 h. Thereafter, the parasites were washed, collected, filtered for purification, and lysed in radioimmunoprecipitation assay (RIPA) buffer for affinity purification. The corresponding strains without biotin treatment were used as controls. The lysate was centrifuged at $12,000 \times g$ for 15 min, and supernatants were incubated with streptavidin magnetic beads (Beaver Biosciences Inc, China) for 6 h at 4°C with gentle shaking. The beads were then washed three times with RIPA lysis buffer, and the proteins bound to the beads were then eluted and solubilized in 1× SDS loading buffer containing 20 mM dithiothreitol. Control samples were processed in the same manner.

**qPCR test.** RNA was extracted from parasite lines using the cell/tissue total RNA isolation kit and then used for cDNA synthesis via HiScript II Q Select RT supermix for quantitative PCR (qPCR) (Vazyme Biotech, China). The resulting cDNA samples were then subjected to qPCR with ChamQ Universal SYBR qPCR master mix (Vazyme Biotech, China), and TgActin was used as a reference gene.

**Immunoprecipitation.** The constructed pEGFP-TgAhp1 and pCMV-TgUrm1 vectors were cotransfected with HEK293T cells, and the medium was discarded after 24 h. We added 400 $\mu$m Dia oxidant to medium for 10 min, and we washed it twice with precooled PBS. Cells were lysed with RIPA and verified by coimmunoprecipitation (co-IP) using anti-green fluorescent protein (GFP) nanobody Magarose beads.

**Yeast two-hybrid assay.** A yeast two-hybrid assay (Y2H) was performed using the Matchmaker GAL4-based two-hybrid system as described in the *Yeast Protocols Handbook* (40; Clontech). The full-length cDNA sequences of TgAhp1 and TgUrm1 were cloned into the pGADT7 and pGBKT7 vectors, and the prey and bait constructs were cotransformed into the yeast strain Y2HGold by the lithium acetate method. The interaction of each cotransformation combination was verified by growing the

cotransformants on minimal −Leu/−Trp (SD-2) medium and −Leu/−Trp/−His/−adenine (SD-4) medium containing 20 mg/mL X-gal.

**Detection of ROS level in Urm1 mutant strains.** Tachyzoites were labeled with 10 $\mu$ mol/L DCFH-DA diluted with Dulbecco's modified Eagle medium (DMEM) at 37°C and 5% $CO_2$ atmosphere for 20 min and then washed three times with PBS. They were treated with 400 $\mu$m Dia oxidant for 10 min or $10^{-4}$ M tBOOH for 2 h and washed three times with PBS. We collected tachyzoites and adjusted the number to be consistent. Total fluorescence intensity of tachyzoites was determined with a fluorescence microplate reader (SpectraMax, USA) with 488 nm of excitation wavelength and 525 nm of emission wavelength.

**Statistical analysis.** Statistical comparisons were performed in Prism 5 (GraphPad Software Inc., USA) using Student's *t* test and Gehan-Breslow-Wilcoxon test. Statistical data were expressed as the mean value of the standard error of the mean (SEM). *P* values of <0.05 and <0.01 were considered statistically significant and extremely significant, respectively.

## ACKNOWLEDGMENTS

This work was supported by the Youth Project of the National Natural Science Foundation of China (grant number 31902289) and the project (grant number ZR2019BC045) supported by Shandong Provincial Natural Science Foundation.

We declare that there are no conflicts of interest regarding the publication of this article.

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
