## [Reviewer comments · Microbiology Spectrum]

Microbiology Spectrum

Involvement of Urm1, a ubiquitin-like protein, in the regulation of oxidative stress response of *Toxoplasma gondii*

Qianqian Tan, Jinwen Wang, Junpeng Chen, Xiaomei Liu, Xiao Chen, Qianqian Xiao, Jinxuan Li, Hongmei Li, Xiaomin Zhao, and Xiao Zhang

Corresponding Author(s): Xiao Zhang, Shandong Agricultural University

Review Timeline:

Submission Date:	November 25, 2021
Editorial Decision:	December 30, 2021
Revision Received:	January 28, 2022
Editorial Decision:	February 17, 2022
Revision Received:	February 22, 2022
Editorial Decision:	March 2, 2022
Revision Received:	March 2, 2022
Accepted:	March 7, 2022

Editor: Björn Kafsack

Reviewer(s): The reviewers have opted to remain anonymous.

Transaction Report:

DOI: <https://doi.org/10.1128/spectrum.02394-21>

December 30, 2021

Prof. Xiao Zhang
Shandong Agricultural University
Taian
China

Re: Spectrum02394-21 (Involvement of Urm1, a ubiquitin-like protein, in the regulation of oxidative stress response of *Toxoplasma gondii*)

Dear Prof. Xiao Zhang:

Both reviewers felt that your manuscript offers important new insights into the role of Urm1 in *Toxoplasma*.

However, both felt that the manuscript contains claims that are not sufficiently supported by the data and that it is missing important details of rationale and methodology.

Additionally, both reviewer felt that the manuscript is in need of professional editing to improve its readability.

I strongly suggest that you address their concerns thoroughly for resubmission.

Link Not Available

Sincerely,

Björn Kafsack

Journals Department
Reviewer comments:

Reviewer #1 (Comments for the Author):

This study is focused on understanding the role of a ubiquitin-like protein, Urm1 in the biology of the *Toxoplasma gondii*.

This is indeed a novel protein and we do not know the precise function of this protein in parasite pathogenesis. However, this study is not well designed, lacks important controls and is poorly written with many of the important rationale and procedural details missing from the manuscript. Further there are grammatical errors in many parts of the text. My comments are as follows.

1. An important control strain that needs to be included to draw any inference is the complemented strain (Where the gene TgURM1 is put back in to the knockout strain and is able to restore the wildtype phenotype). Without this strain one cannot make any conclusions about the role of URM1.
2. Where is the western blot showing size of this protein?
3. In Fig 2C, how the invasion assay was performed? How the invasion efficiency was determined?
4. In Fig 2D, how was the replication assay was done? If the added parasites were washed after 30 min of invasion, according to the data shown, it looks like mutant parasites have defect in replication. Authors cannot conclude there is no defect in endodyogeny by using just using two markers GAP45 and IMC1 which both are markers of inner membrane complex (Fig 2H).
5. Fig 2F, how was the motility ability determined? How it was quantified? Is there a difference in the three types of motility (circular, helical and gliding)?
6. Fig 2G, how the egress assay was done and quantified?
7. The authors are working RH strain that does not form cysts. Their conclusion that the mice inoculated with the knockout strain survive because they fail to differentiate does not make any sense.
8. TgURM localizes to the parasite cytoplasm. There is no rationale provided for expression of this protein in HEK293T cells.
9. Authors are trying to identify interactors of TgURM using different approaches in sections 3.5 and 3.6. Where is the mass spec data of interactors identified by proximity-based labeling? Why there were no common proteins identified by different approaches? How then can one trust the data they have?
10. Sections 3.6 and 3.7 are very poorly written and very confusing. The authors do not provide any logical rationale why they are doing these experiments.
11. Section 3.7, line 306, which study you are referring to?
A few examples where sentences need to be corrected are as follows.

Lines 29-30

Lines 80-81

Reviewer #2 (Comments for the Author):

This manuscript studied a UBL-like protein in a protozoan parasite, *Toxoplasma gondii*. The authors generated a straight knockout in the study and identified the defects on invasion, replication, and acute virulence in the knockout. Upon oxidative stress, the parasites perform urmylation by conjugating TgUrm1 to other proteins in the parasites. This work characterized a novel subcellular process in *Toxoplasma*. However, there are a few issues the authors need to address to strengthen this work. I listed them below.

Major:

1. It is unclear how TgUrm1 is tagged with 3xHA at its N-terminus in Fig. 1C. Did the Cas9 protein cut the internal region of the TgUrm1 gene for tagging? Which promoter was the tagged TgUrm1 driven?
2. I suggest the authors use the standard survival plot to document the virulence assay. The significance can be calculated using the appropriate statistical analysis.
3. In Fig. 3, the RH Δ ku80 strain used in this study belongs to Type 1, which shows less tendency for cyst conversion than Type 2 strain. Therefore, the BAG1 expression did not increase significantly post alkali induction, as the authors showed in this study. Therefore, I don't feel confident that the TgUrm1 knockout displayed defects in cyst formation.
4. In Fig. 4F, the TgUrm1 knockout pretreated with diamide killed mice at the same percentage compared to non-treated TgUrm1 knockout. The authors need to run statistical analysis before concluding that oxidative stress harms the pathogenesis of the TgUrm1 knockout.
5. In Fig. 7F, a similar concern to item 4, the tBOOH-treated TgUrm1 knockout did not show virulence loss compared to non-treated TgUrm1 knockout.

6. In Fig. 4B, tBOOH did not trigger significant urmylation relative to non-treated parasites. Although both proteins interact with each other upon the presence of oxidative stress, how can the interaction be linked to urmylation?

7. If the TgUrm1 is involved in regulating the stress response, the TgUrm1 knockout would display higher ROS/RNS levels in the presence of oxidative stress than WT parasites, maybe even without oxidative stress. The authors may consider measuring the ROS/RNS levels.

Minor:

1. Line 24: "deficiency mutations", remove the deficiency
2. Line 182: "growth deficiency" should be growth defect
3. Lacking scale bar in IFA pictures
4. Line 207: Define Urm1-OE, overexpression of TgUrm1 in the knockout?
5. Line 210, sera were tested, not detected
6. Need scale bar for all IFA images

Staff Comments:

Preparing Revision Guidelines

Please return the manuscript within 60 days; if you cannot complete the modification within this time period, please contact me. If you do not wish to modify the manuscript and prefer to submit it to another journal, please notify me of your decision immediately so that the manuscript may be formally withdrawn from consideration by Microbiology Spectrum.

Dear editor:

I have revised the manuscript based on the comments of the reviewers, and the details are as follows:

Reviewer #1 (Comments for the Author):

1. An important control strain that needs to be included to draw any inference is the complemented strain (Where the gene TgURM1 is put back in to the knockout strain and is able to restore the wildtype phenotype). Without this strain one cannot make any conclusions about the role of URM1.

Lacking of an intron in Urm1 gene, the strain in which the original TgUrm1 locus was replaced with the TgUrm1 tagged with 3xHA at TgUrm1's N-terminus, was used as the complemented strain. The complemented strain still used the endogenous promoter for transcription initiation. We have added supplementary data, including plaque, invasion, replication, virulence of complemented strain in Figure 2B-D, Figure 3A-B, and Figure 7B-G.

2. Where is the western blot showing size of this protein?

Western-blot result was showed in Figure 1C, and the molecular weight of HA tagged TgURM1 protein was 28 KD.

3. In Fig 2C, how the invasion assay was performed? How the invasion efficiency was determined?

Protocol of invasion assay and the calculation method for invasion efficiency determination were showed on lines 108-118.

4. In Fig 2D, how was the replication assay was done? If the added parasites were washed after 30 min of invasion, according to the data shown, it looks like mutant parasites have defect in replication. Authors cannot conclude there is no defect in endodyogeny by using just using two markers GAP45 and IMC1 which both are markers of inner membrane complex (Fig 2H).

Protocol of replication assay and the calculation method were showed on lines 119-128. Knockout strain does have replication defects in Figure 2D. Of course, our experimental results only showed that knocking out URM1 had no effect on the formation of IMC1 in progeny parasites, but it was hard to prove the relevance of knocking out URM1 on other replication processes of parasites. In addition, some sentences in the manuscript were inaccurate. In order to avoid misleading, we have deleted this part.

5. Fig 2F, how was the motility ability determined? How it was quantified? Is there a difference in the three types of motility (circular, helical and gliding)?

In general, gliding types of *T. gondii* includes circular gliding, upright twirling, and helical rotation circular. Parasites that could not complete one of the three gliding modes were defined as

gliding defect. Protocol of gliding assay and the calculation method were showed on lines 129-138.

6. Fig 2G, how the egress assay was done and quantified?

Protocol of egress assay and the calculation method were showed on lines 139-144.

7. The authors are working RH strain that does not form cysts. Their conclusion that the mice inoculated with the knockout strain survive because they fail to differentiate does not make any sense.

In general, the RH strain does not form cysts, but with some special inducer or knocking out some special genes, the RH strain still has a tendency to form cysts. Of course, our experimental results could not fully prove the relevance of TgURM1 and cyst formation, and some sentences in the manuscript were inaccurate. In order to avoid misleading, we have deleted this part.

8. TgURM1 localizes to the parasite cytoplasm. There is no rationale provided for expression of this protein in HEK293T cells.

Immunofluorescence result of TgUrm1 location in HEK293T was added in Figure 4A.

9. Authors are trying to identify interactors of TgURM using different approaches in sections 3.5 and 3.6. Where is the mass spec data of interactors identified by proximity-based labeling? Why there were no common proteins identified by different approaches? How then can one trust the data they have?

A total of 51 proteins were identified using biotin-adjacent labeling method in the presence of Dia, while 107 proteins were identified by proteomics. Comprehensive analysis of all identification, 22 promising proteins identified by both methods were selected (Fig. 5E), which involved in oxidative stress response, immune evasion and parasite metabolism (lines 275-279).

10. Sections 3.6 and 3.7 are very poorly written and very confusing. The authors do not provide any logical rationale why they are doing these experiments.

TgAHP1 (TG GT1_286630) was first reported in our previous study “Alkyl Hydroperoxide Reductase as a Determinant of Parasite Antiperoxide Response in *Toxoplasma gondii*”, which was identified as a peroxisomal protein with cysteine-dependent peroxidase activity in *T. gondii*. Interestingly, TgAHP1 was found in the mass spectrum data (Figure 5E), indicating that AHP1-URM1 complexes may be detected in the presence of oxidant (lines 281-284).

We have changed some sentences and cited this reference to make the manuscript more logical. In addition, we have rewritten the Sections 3.6 and Sections 3.7 to make the manuscript more logical and complete.

11. Section 3.7, line 306, which study you are referring to?

This section referring to our previous study "Alkyl hydroperoxide reductase as a determinant of parasite antiperoxide response in *Toxoplasma gondii*", and we have quoted this reference.

We have rewritten the Section 3.7 to make the manuscript more logical.

A few examples where sentences need to be corrected are as follows.

Lines 29-30

We have changed this sentence to make it more logical (lines 29-30).

Lines 80-81

We have changed this sentence to make it more logical (lines 88-90).

Reviewer #2 (Comments for the Author):

1. It is unclear how TgUrm1 is tagged with 3xHA at its N-terminus in Fig. 1C. Did the Cas9 protein cut the internal region of the TgUrm1 gene for tagging? Which promoter was the tagged TgUrm1 driven?

Lacking of an intron in Urm1 gene, the strain in which the original TgUrm1 locus was replaced with the TgUrm1 tagged with 3xHA at TgUrm1's N-terminus (lines 205-207). The HA tagged TgUrm1 strain still used the endogenous promoter for transcription initiation, and the Cas9 protein cut the internal region of the TgUrm1 gene for tagging.

2. I suggest the authors use the standard survival plot to document the virulence assay. The significance can be calculated using the appropriate statistical analysis.

Thank you very much for the reviewer's suggestion. We have used the standard survival plot to document the virulence assay and the significance was analyzed using Gehan-Breslow-Wilcoxon tests, referring to the reference "Functional analysis of *Toxoplasma* lactate dehydrogenases suggests critical roles of lactate fermentation for parasite growth in vivo"

3. In Fig. 3, the RH Δ ku80 strain used in this study belongs to Type 1, which shows less tendency for cyst conversion than Type 2 strain. Therefore, the BAG1 expression did not increase significantly post alkali induction, as the authors showed in this study. Therefore, I don't feel confident that the TgUrm1 knockout displayed defects in cyst formation.

In general, the RH strain does not form cysts, but with some special inducer or knocking out some special genes, the RH strain still has a tendency to form cysts. Of course, our experimental results could not fully prove the relevance of TgURM1 and cyst formation, and some sentences in the manuscript were inaccurate. In order to avoid misleading, we have deleted this part.

4. In Fig. 4E, the TgUrm1 knockout pretreated with diamide killed mice at the same percentage compared to non-treated TgUrm1 knockout. The authors need to run statistical analysis before concluding that oxidative stress harms the pathogenesis of the TgUrm1

knockout.

The results have been removed to Figure 7F to make the manuscript more logical, and Gehan-Breslow-Wilcoxon tests was used to analyse the significance of virulence.

5. In Fig. 7F, a similar concern to item 4, the tBOOH-treated TgUrm1 knockout did not show virulence loss compared to non-treated TgUrm1 knockout.

Gehan-Breslow-Wilcoxon tests was used to analyse the significance of virulence between the two strains in Figure 7F.

6. In Fig. 4B, tBOOH did not trigger significant urmylation relative to non-treated parasites. Although both proteins interact with each other upon the presence of oxidative stress, how can the interaction be linked to urmylation?

In the previous Fig 4B, tBOOH failed to trigger significant urmylation, which was due to the using of low concentration tBOOH. To make the manuscript more logical, tBOOH-induced ubiquitination was re-performed using the unified concentration (10^{-4} M tBOOH) of Figure 7, and supplementary result was showed in Figure 7A, which indicated that the treatment with 10^{-4} M tBOOH also triggered significant Urm1 urmylation. Both Figure 4B and 4C in the original manuscript actually indicated that multiple oxidants could induce TURM1 ubiquitination. Thus, we removed the Figure 4B.

7. If the TgUrm1 is involved in regulating the stress response, the TgUrm1 knockout would display higher ROS/RNS levels in the presence of oxidative stress than WT parasites, maybe even without oxidative stress. The authors may consider measuring the ROS/RNS levels.

Supplementary experiments for ROS level detection were performed, and the results were added on lines 302-306 and Figure 7G.

Minor:

1. Line 24: "deficiency mutations", remove the deficiency

We have removed the deficiency on line 24.

2. Line 182: "growth deficiency" should be growth defect

We have changed "growth deficiency" to "growth defect" on line 220.

3. Lacking scale bar in IFA pictures

We have added the scale bar in all IFA pictures, including Figure 1D, Figure 2E, Figure 4A, Figure 5C.

4. Line 207: Define Urm1-OE, overexpression of TgUrm1 in the knockout?

Yes, Urm1-OE is a strain, which TgUrm1 overexpressed in the knockout strain. We have

defined the strain on lines 230-231.

5. Line 210, sera were tested, not detected

We have changed " detected " to " tested " on lines 233.

6. Need scale bar for all IFA images.

We have added the scale bar in IFA pictures, including Figure 1D, Figure 2E, Figure 4A, Figure 5C.

February 17, 2022

Prof. Xiao Zhang
Shandong Agricultural University
Taian
China

Re: Spectrum02394-21R1 (Involvement of Urm1, a ubiquitin-like protein, in the regulation of oxidative stress response of *Toxoplasma gondii*)

Dear Prof. Xiao Zhang:

Thank you for submitting your manuscript to Microbiology Spectrum. As you will see your paper is very close to acceptance. Please modify the manuscript along the lines suggested by the reviewers. As these revisions are quite minor, I expect that you should be able to turn in the revised paper in less than 30 days, if not sooner. If your manuscript was reviewed, you will find the reviewers' comments below.

When submitting the revised version of your paper, please provide (1) point-by-point responses to the issues I raised in your cover letter, and (2) a PDF file that indicates the changes from the original submission (by highlighting or underlining the changes) as file type "Marked Up Manuscript - For Review Only". Please use this link to submit your revised manuscript. Detailed instructions on submitting your revised paper are below.

Link Not Available

Sincerely,

Björn Kafsack

Reviewer comments:

Reviewer #1 (Comments for the Author):

1. Authors have not included any details of how the complemented strain was generated either in the materials and methods or the results section.
2. In Figure 2C and 2D, they have included the "complemented" strain but they not describe the results obtained with this strain in the results section. Authors do not provide reasons why the complemented strain was not included in the motility and egress assays shown in Fig 2E, F and G.
3. Line 29 should be...In conclusion, TgURM1 is a UBL protein..
4. Line 89 should be....TgAHP1 is an...

Reviewer #2 (Comments for the Author):

The authors addressed some questions from both reviewers. I have some concerns listed below. Please explain them.

1. For Q1 from reviewer 1, the authors mentioned that they removed the first intron and complemented the TgUrm1 gene in the original locus of TgUrm1. Please describe details for the creation of the knockout and complementation strains. It would be great

to include a detailed schematic illustration. The current one in Fig 2A is too simple.

2. The authors need to provide more experimental details for their phenotypic characterizations.

3. Fig. 4A is inconclusive. I cannot tell URM1 is in the cytoplasm.

4. For Q1 from reviewer 2, please refer to the first question. Need more details.

5. For Q5 from reviewer 2, I am unsure if you can get biological significance among these very mild virulence differences using five mice.

6. For Q7 from reviewer 2, in Fig 7G, you need to include non-treated WT, knockout, and complementation strains as references. Otherwise, the treatment data are meaningless.

Overall, the authors deleted a few original observations from the manuscript during the revision, which weakens the significance of this study.

Preparing Revision Guidelines

- point-by-point responses to the issues I raised in your cover letter
- Upload a compare copy of the manuscript (without figures) as a "Marked-Up Manuscript" file.
- Each figure must be uploaded as a separate file, and any multipanel figures must be assembled into one file.
- Manuscript: A .DOC version of the revised manuscript
- Figures: Editable, high-resolution, individual figure files are required at revision, TIFF or EPS files are preferred

Please return the manuscript within 60 days; if you cannot complete the modification within this time period, please contact me. If you do not wish to modify the manuscript and prefer to submit it to another journal, please notify me of your decision immediately so that the manuscript may be formally withdrawn from consideration by Microbiology Spectrum.

Dear editor:

We have revised the manuscript based on the comments of the reviewers, and the details are as follows:

Reviewer #1 (Comments for the Author):

1. Authors have not included any details of how the complemented strain was generated either in the materials and methods or the results section.

The details of the generation of Urm1 knockout and complemented strains were shown in the **Materials and Methods** on lines 102-112.

2. In Figure 2C and 2D, they have included the "complemented" strain but they not describe the results obtained with this strain in the results section. Authors do not provide reasons why the complemented strain was not included in the motility and egress assays shown in Fig 2E, F and G.

A supplemented description " The growth of defect was further confirmed, and the attenuated invasion and proliferation trends were observed in Δ Urm1 strain, comparing to wild-type RH Δ Ku80 strain and Com Δ Urm1 strain " was added on lines 233-235 to describe the results of "complemented" strain.

There was no significant difference in the motility and egress process between the "complemented" strain and wild-type strain, and we neglected to provide the data of the "complemented" strain. We have supplemented the data of complemented strain in Fig 2E-G.

3. Line 29 should be...In conclusion, TgURM1 is a UBL protein..

We have changed " ...In conclusion, TgURM1 as a UBL protein. " to " ...In conclusion, TgURM1 is a UBL protein. " on line 29.

4. Line 89 should be....TgAHP1 is an...

We have changed "TgAHP1 as an... " to "....TgAHP1 is an... " on line 89.

Reviewer #2 (Comments for the Author):

1. For Q1 from reviewer 1, the authors mentioned that they removed the first intron and complemented the TgUrm1 gene in the original locus of TgUrm1. Please describe details for the creation of the knockout and complementation strains. It would be great to include a detailed schematic illustration. The current one in Fig 2A is too simple.

The details of the generation of Urm1 knockout and complemented strains were shown in the

Materials and Methods on lines 102-112, and the schematic illustration was shown in Fig 2A.

2. The authors need to provide more experimental details for their phenotypic characterizations.

Alignment details for the evolutionary tree were added on lines 492-500.

3. Fig. 4A is inconclusive. I cannot tell URM1 is in the cytoplasm.

Cytoplasmic separation was made in URM1 transfected cells, and the Western blotting was performed to confirm the cytoplasmic localization of URM1 on lines 262-264. The result was shown in Fig.4B.

4. For Q1 from reviewer 2, please refer to the first question. Need more details.

The details of the generation of Urm1 knockout and complemented strains were shown in the **Materials and Methods** on lines 102-112.

5. For Q5 from reviewer 2, I am unsure if you can get biological significance among these very mild virulence differences using five mice.

Three independent replicates were made in *T. gondii* pathogenicity test with five mice/group each time. Three independent replicate results were combined and re-analyzed for the significance of virulence in Fig 7F.

6. For Q7 from reviewer 2, in Fig 7G, you need to include non-treated WT, knockout, and complementation strains as references. Otherwise, the treatment data are meaningless.

In Fig 7G, the increase rate of ROS in different strains was calculated. The formula for the ROS increase rate is simplified to the following: Increase rate= $[OX(+)-OX(-)]/OX(-)\times 100\%$.

Where "OX(+)" is the ROS levels of oxidant treatment group, "OX(-)" is the ROS levels of oxidant non-treatment group. The increase rates actually included the all strains (treated group or non-treated). The supplement was added on lines 577-580.

March 2, 2022

Prof. Xiao Zhang
Shandong Agricultural University
Taian
China

Re: Spectrum02394-21R2 (Involvement of Urm1, a ubiquitin-like protein, in the regulation of oxidative stress response of *Toxoplasma gondii*)

Dear Prof. Xiao Zhang:

Thank you for submitting your manuscript to Microbiology Spectrum. As you will see your paper is very close to acceptance. Please modify the manuscript along the lines I have recommended. As these revisions are quite minor, I expect that you should be able to turn in the revised paper in less than 30 days, if not sooner. If your manuscript was reviewed, you will find the reviewers' comments below.

When submitting the revised version of your paper, please provide (1) point-by-point responses to the issues I raised in your cover letter, and (2) a PDF file that indicates the changes from the original submission (by highlighting or underlining the changes) as file type "Marked Up Manuscript - For Review Only". Please use this link to submit your revised manuscript. Detailed instructions on submitting your revised paper are below.

Link Not Available

Sincerely,

Björn Kafsack

Reviewer comments:

Preparing Revision Guidelines

- point-by-point responses to the issues I raised in your cover letter
- Upload a compare copy of the manuscript (without figures) as a "Marked-Up Manuscript" file.
- Each figure must be uploaded as a separate file, and any multipanel figures must be assembled into one file.
- Manuscript: A .DOC version of the revised manuscript
- Figures: Editable, high-resolution, individual figure files are required at revision, TIFF or EPS files are preferred

Please return the manuscript within 60 days; if you cannot complete the modification within this time period, please contact me. If you do not wish to modify the manuscript and prefer to submit it to another journal, please notify me of your decision immediately so that the manuscript may be formally withdrawn from consideration by Microbiology Spectrum.

Dear editor:

We have revised the manuscript based on the comments.

March 3, 2022

Prof. Xiao Zhang
Shandong Agricultural University
Taian
China

Re: Spectrum02394-21R3 (Involvement of Urm1, a ubiquitin-like protein, in the regulation of oxidative stress response of *Toxoplasma gondii*)

Dear Prof. Xiao Zhang:

Your manuscript has been accepted, and I am forwarding it to the ASM Journals Department for publication. You will be notified when your proofs are ready to be viewed.

Sincerely,

Björn Kafsack
Editor, Microbiology Spectrum
